# Age-Related Distinctions in EEG Signals during Execution of Motor Tasks Characterized in Terms of Long-Range Correlations

**DOI:** 10.3390/s20205843

**Published:** 2020-10-15

**Authors:** Alexey N. Pavlov, Elena N. Pitsik, Nikita S. Frolov, Artem Badarin, Olga N. Pavlova, Alexander E. Hramov

**Affiliations:** 1Saratov State University, Astrakhanskaya Str. 83, 410012 Saratov, Russia; pavlov.lesha@gmail.com (A.N.P.); pavlova_olya@yahoo.com (O.N.P.); 2Neuroscience and Cognitive Technology Laboratory, Center for Technologies in Robotics and Mechatronics Components, Innopolis University, Universitetskaya Str. 1, 420500 Innopolis, Russia; e.pitsik@innopolis.ru (E.N.P.); n.frolov@innopolis.ru (N.S.F.); a.badarin@innopolis.ru (A.B.); 3Lobachevsky University, 23 Gagarina Avenue, 603950 Nizhny Novgorod, Russia; 4Saratov State Medical University, Bolshaya Kazachya Str. 112, 410012 Saratov, Russia

**Keywords:** detrended fluctuation analysis, long-range correlations, electroencephalography (EEG), motor-related brain activity, aging

## Abstract

The problem of revealing age-related distinctions in multichannel electroencephalograms (EEGs) during the execution of motor tasks in young and elderly adults is addressed herein. Based on the detrended fluctuation analysis (DFA), differences in long-range correlations are considered, emphasizing changes in the scaling exponent α. Stronger responses in elderly subjects are confirmed, including the range and rate of increase in α. Unlike elderly subjects, young adults demonstrated about 2.5 times more pronounced differences between motor task responses with the dominant and non-dominant hand. Knowledge of age-related changes in brain electrical activity is important for understanding consequences of healthy aging and distinguishing them from pathological changes associated with brain diseases. Besides diagnosing age-related effects, the potential of DFA can also be used in the field of brain–computer interfaces.

## 1. Introduction

Aging causes changes in the human life, which often include physical and mental impairments, psychological, and social changes. Research on aging is commonly associated with abnormal brain dynamics and the related diseases, such as Alzheimer’s disease and dementia. The physiological mechanisms accompanying the development of these disorders of brain activity have been clearly established [1,2,3,4,5], but some features of pathological dynamics are revealed even in healthy aging [6], and their analysis can provide markers of hidden stages of disorders. Healthy aging alters the neurochemical and structural properties of the brain, that leads to decreased cognitive and motor functions during daily activities in older adults. Age-related neural impairments are quantitatively assessed by a longer reaction time, reduced coordination, and motor control [7,8], which limit the performance of complex motor tasks [9,10,11]. The motor impairments that often occur in elderly adults greatly affect their daily life. Some studies [12,13,14,15] have established additional brain areas involved during the motor activity with aging to overcome structural changes in brain dynamics. This involvement serves as a compensatory mechanism [16,17]. Due to this, execution of motor tasks is expected to differ between younger and elderly subjects.

Age-related impairments of cognitive or motor functions are reflected in the electrical activity of the brain and can be studied based on electroencephalogram (EEG) signals. Such analysis is able to reveal features of brain dynamics in the resting state or during motor/cognitive tasks with various signal processing approaches. In particular, healthy aging increases the complexity of baseline EEGs [18], and this effect is evident in the frontal inferior and sensorimotor areas [19]. Therefore, a number of complexity measures can be applied to identify mild impairments arising, e.g., at the latent stages of neurodegenerative disorders [20,21]. In addition to entropy-based tools that are widely used to quantify complexity, methods dealing with the multifractal organization of EEG signals offer new possibilities [22]. Another set of approaches is related to time-frequency analysis in separate frequency bands, and besides the conventional spectral analysis, it involves more recent techniques based on wavelets [23,24,25], empirical modes [26,27], machine learning [28,29], etc. These approaches provide a clear physiological interpretation of the changes that have occurred in terms of θ (4–8 Hz), α (8–12 Hz), β (15–30 Hz), and other EEG rhythms.

Spectral analysis is closely related to the correlation analysis of experimental data. If ranges of slow-wave activity need to be characterized, a study of long-range correlations in physiological signals is often performed to reveal structural changes that can occur, e.g., due to the transition from normal to pathological dynamics of physiological control systems, changes in environmental conditions, etc. The characterization of long-range correlations has limitations even in the case of stationary dynamics of complex systems, if the correlation function decreases rapidly. When this function approaches zero, it becomes difficult to reliably estimate its power-law behavior, and methods based on the fluctuation analysis are applied, with detrended fluctuation analysis (DFA) [30,31,32] being a commonly used tool that includes a trend removal procedure as a part of an algorithm. DFA can be applied to study the time-varying dynamics of complex systems without preprocessing. Although there is a discussion about the limitations of DFA [33,34,35,36], and the presence of nonstationarity can affect the results, i.e., a preprocessing stage is still required, this method is widely used in physiology due to its simplicity and efficiency for quantifying the power-law statistics of complex processes. The use of DFA in EEG-based studies allows not only to distinguish between different physiological states (normal and pathological brain dynamics, baseline activity and sudden changes due to external influences, etc.), but also to characterize the provoked short-term reactions during motor/cognitive tasks when the datasets under study include transients [37,38]. This is the case for brain–computer interfaces which require rapid recognition of specific EEG patterns in order to transform them into control commands for hardware.

In this study, we examine the DFA’s ability to quantify age-related distinctions in EEG during execution of motor tasks between young and elderly adults. We show how cortical activity differs between these groups and demonstrate stronger changes in correlation properties for elderly adults. Further, we compare distinctions in EEG signals between the motor activity of the dominant and non-dominant hands and report significantly more pronounced differences for the group of young subjects.

The paper is organized as follows. Section 2 is devoted to description of the DFA method used for data processing. In Section 3, we consider the experimental procedures in the two age groups of subjects. Our main results describing age-related distinctions in the electrical activity of the brain when performing motor tasks are presented in Section 4. In Section 5 we discuss inter-group differences and give our assumptions about their reasons. Section 6 briefly summarizes the findings of the study.

## 2. DFA Method Used for EEG Data Processing

DFA was proposed by Peng et al. [30,31] as an approach to correlation analysis of physiological datasets. It is a variant of the root mean square (RMS) analysis of random walks and includes the fitting of slow nonstationarity, treated as a trend, with further characterization of fluctuations around the detrended signal profile. The algorithm consists of the following steps.

(1)Construction a profile y(k) of a signal x(i), i=1,…,N, k=1,…,N (or random walk in terms of the theory of random processes).
(1)$$y\left(k\right)=\sum_{i=1}^{k}\left[x(i)-\langle x\rangle \right],\;\langle x\rangle =\sum_{i=1}^{N}x\left(i\right).$$(2)Dividing the profile y(k) into non-overlapping segments of length *n* (*n* < *N*).(3)Fitting the trend $y_{n}\left(k\right)$ within each segment with the least-squares method.(4)Computing RMS deviation of y(k) from $y_{n}\left(k\right)$.
(2)$$F\left(n\right)=\sqrt{\frac{1}{N}\sum_{k=1}^{N}{\left[y\left(k\right)-y_{n}\left(k\right)\right]}^{2}.}$$(5)Repeating steps 2–4 for different values of *n* to obtain an increasing dependence F(n).(6)Estimation of the scaling exponent α
(3)$$F\left(n\right)∼n^{\alpha }$$
by representing Equation (3) in the log-log plot and linear fitting in the selected range of *n*. Such power-law behavior is typical for many stochastic processes, although α can vary with *n* for inhomogeneous data sets.

Thus, Figure 1 shows an example of the dependence lg F vs. lg n for an EEG signal, which illustrates a nearly power-law behavior characterized by the scaling exponent α≈1.03 in the range lg n>1.2. The slope of such relationship can differ depending on the time scale. This is a fairly typical case for physiological signals, where correlation properties are often different for short- and long-range correlations, and multiscale behavior usually occurs [31]. Analysis of long-range correlations is performed to reveal pathological changes in physiological systems, as various failures lead to a breakdown of correlation behavior for large time scales, and the exponent α can serve as a marker of such changes. In particular, the analysis of long-range correlations associated with large lg n has the potential for bedside and ambulatory monitoring in cardiology [31]. It can also identify specific changes in the slow-wave dynamics of electrical activity in the brain associated with various conditions [39] that motivate the use of DFA in the current study. In contrast to direct estimation of the correlation function, which approaches zero with increasing time lag and complicates the analysis of its scaling features for noisy data sets, DFA is better adapted to characterize power-law behavior in this range of scales.

At step 3, a piecewise linear fitting is typically applied as the simplest and most universal approach, but the method has no restrictions with respect to other types of fitting. The α exponent quantifies anti-correlations in the signal x(i), when small and large values tend to alternate (α<1/2), uncorrelated behavior (α=1/2), positive power-law correlations (1/2<α<1), positive correlations with other (not power-law) behavior (α>1). Currently, DFA is used to identify correlations in the dynamics of systems with both stable and time-varying parameters. Thus, even for stationary processes, DFA can outperform the standard correlation analysis when considering long-range correlations. In contrast, short-range correlations are better characterized by the correlation function. Some DFA applications can be found, e.g., in [39,40,41,42,43,44,45]. In contrast to the standard approach, here we introduce a floating window into DFA for evaluating α as a function of time. This allows us to characterize both the variations in α(t) and the rate of these variations, which can serve as an additional diagnostic marker.

## 3. Experiment

### 3.1. Subjects

Experiments were carried out on two groups of healthy volunteers (Russians): 10 young adults (3 females and 7 males; age 26.1±5.15 (mean ± SD); range: 19–33) and 10 elderly adults (6 females and 4 males; age 65 ± 5.69 (mean ± SD); range: 55–72). All volunteers were right-handed and had no history of neural pathologies (brain tumors, trauma, or stroke-related medical conditions). Before participation, they signed a written informed consent and were pre-informed about the goals of the study and experimental procedures. The experiments were performed in accordance with the Declaration of Helsinki. The protocol was approved by the local Ethics Committee of the Innopolis University (Kazan, Russia).

### 3.2. Experimental Procedures

The volunteers sat on a chair with the hands placed comfortably on the table desk in front of them, palms up. Before executing motor tasks, a background EEG was acquired for 5 min when the volunteers were in a relaxed state, with their eyes open, and were not focusing on any specific thoughts. Then, a repetitive series of 60 motor tasks was carried out (30 tasks per hand). Each task consisted of clenching the hand into a fist after the audio signal and holding it in a clenched state until the repeated signal. The type of movement depended on the duration of the signal: a short beep (0.3 s) was used for non-dominant hand (left hand, LH) movements, and a longer audio signal (0.75 s) was applied for dominant hand (right hand, RH) movements. In contrast with a widely applied visual-pacing of motor actions [46], audio commands induce greater cortical activation related with much more pronounced launching of perception–action loops associated with sensorimotor integration and affected by healthy aging [47,48]. The order of tasks (left of right hand) was chosen randomly to avoid training effects. Each individual motor task included the baseline EEG measurements (2 s), then the active part of the task (clenching the hand and holding it in a clenched state, 4–5 s) and an unclenching the hand, followed by a prolonged pause (6–8 s) as shown in Figure 2.

### 3.3. EEG Data Acquisition and Preprocessing

Multichannel EEG signals were acquired with a sampling rate of 250 Hz using an “Encephalan-EEG-19/26” electroencephalograph (Medicom MTD company, Taganrog, Russian Federation) with 31 electrodes (O2, O1, P4, P3, C4, C3, F4, F3, Fp2, Fp1, P8, P7, T8, T7, F8, F7, Oz, Pz, Cz, Fz, Fpz, FT7, FC3, FCz, FC4, FT8, TP7, CP3, CPz, CP4, TP8) located in accordance with the “10–10” registration scheme, two reference electrodes A1 and A2 on the earlobes, and a ground electrode N just above the forehead. To acquire the EEG signals we used the cup adhesive Ag/AgCl electrodes placed on the “Tien–20” paste (Weaver and Company, Denver, CO, USA). Immediately before the experiments started, we performed all necessary procedures to increase the conductivity of the participant’s skin and reduce its resistance using the abrasive NuPrep gel (Weaver and Company, Denver, CO, USA). After the electrodes had been installed, the impedance was monitored throughout the experiments. We kept the impedance values close to a 2/5 kΩ interval. The electroencephalograph has a registration certificate of the Federal Service for Supervision in Healthcare No. FCP 2007/00124 of 7 November 2014 and a European certificate CE 538571 of the British Standards Institute. Data filtering was carried out using a Butterworth bandpass filter with cut-off frequencies of 1 Hz and 100 Hz and using a 50 Hz notch filter. Artifacts caused by eye blinking and heartbeats were removed using an approach based on the independent component analysis [49].

For further analysis, fragments of multichannel EEG data associated with each individual motor task were centered at the beginning of the first audio signal. In addition, visual control of these fragments was provided to exclude fragments corrupted by artifacts that were not removed after application of the automatic artifact suppression algorithm. As a result of such visual control, half of the fragments (15 motor tasks per hand) were selected being less distorted by artifacts.

All this preprocessing was done using the MNE package for Python 3.7 (ver. 0.20.0) [50]. The analyzed EEG data are available online [51].

## 4. Results

Our previous studies have revealed the ability to recognize movement types, including mental intentions, based on the scaling exponent α of long-range correlations that can be used, e.g., for brain–computer interfaces [38]. With these findings, here we performed a windowed DFA within a 2 s floating window (500 samples). For each participant, the dependencies of α on the position of the window were estimated for every EEG fragment related to an individual motor task and averaged over all fragments. Then, intra-group averaging was done for young and elderly adults. The results are shown in Figure 1 as mean values ± SE for the C4 EEG channel.

The variability of α is quite strong within each group. This circumstance does not allow directly using the absolute values of α as a diagnostic marker of age-related changes. Despite this circumstance, there is a tendency for the scaling exponent to increase when participants clench a hand into a fist or unclench a fist. As the estimates are performed within a floating window of 2 s duration, the dependencies in Figure 3 are pretty smooth. Their visual inspection shows that the range and rate of increase in α may differ between the groups of young and elderly adults. In order to better recognize and quantify these distinctions, we will further consider the normalized dependencies α(t). For this purpose, we select two parts related to the most pronounced changes (i) during clenching the hand into a fist and (ii) during unclenching the fist, and normalize the related α(t) to the α values corresponding to the starting point of each part. Therefore, for every individual motor task, we consider a grows of α in normalized units, starting with α = 1.0 for both movements—clenching and unclenching a fist, and then we carry out a double averaging: over repetitive tasks for an individual participant and for the whole group. In addition, we compare the results of young and elderly adults for all motor tasks (i.e., LH and RH movements), and for each hand separately. At this stage, we discuss the results for two EEG channels (C3 and C4) that are both closely related to left-right relative tasks.

Figure 4 illustrates inter-group difference without dividing motor tasks for dominant and non-dominant hands (mean values of normalized α ± SE). This figure clearly shows the distinctions in the rate of increase in α, i.e., in the slopes *r* of these dependencies (Table 1),
(4)$$r=\frac{d\alpha }{dt}.$$

For both parts of the motor tasks, significant differences were found according to the Mann–Whitney test for the C4 channel (*p* < 0.05), while for the C3 channels the distinctions are less pronounced. Elderly adults tend to demonstrate stronger responses in electrical activity of the brain, reflected in a wider range of α changes and faster α growth.

To understand whether the choice of the dominant or non-dominant hand affects the response, we show in Figure 5 and Figure 6 the results of statistical analysis for each hand separately. Thus, Figure 5 provides a comparison of responses to motor tasks performed by the dominant hand (RH). Although the behavior is similar to Figure 4, the differences become less pronounced for both channels (Table 1), but the C4 channel provides a clearer inter-group separation.

In the case of the non-dominant hand (LH), the inter-group separation is better (Figure 6, Table 1). This allows us to conclude that, for the selected EEG channel, the motor tasks carried out by the non-dominant hand lead to stronger distinctions in the electrical activity of the brain between young and elderly adults. Again, the differences for the C4 channel outperform those observed for the C3 channel.

Let us now consider how the revealed phenomena depend on the position of the electrode. The absolute values of *r* vary between subjects and channels. To compare the effects of a dominant or non-dominant hand, we estimated the difference between the *r* values for LH and RH, and then averaged it across all channels and entire groups. The resulting mean differences take the following values (mean ± SE): 0.0107 ± 0.0012 for elderly adults and 0.0259 ± 0.0043 for young adults, confirming that young participants demonstrate significantly stronger distinctions in responses to dominant or non-dominant hand movements unlike elderly subjects. Typically, motor tasks executed by LH and RH in the elderly adults cause comparable changes in long-range correlations of EEG datasets, while in young adults the reactions to motor tasks from the non-dominant hand are weaker.

Figure 7 shows how the average difference between the *r* values for elderly and young participants is distributed across EEG channels. The strongest distinctions are associated with the area of sensorimotor cortex (Cz, C4). By analogy with the previous findings, motor tasks performed by the non-dominant hand (LH) caused the clearest inter-group separation.

## 5. Discussion

We examined how motor tasks performed by young and elderly adults affect the correlation features of multichannel EEG recordings. For this purpose, we used DFA to quantify the scaling exponent of long-range power-law correlations in electrical activity of the brain, when volunteers execute fine motor tasks, consisting of clenching/unclenching a hand into a fist. The phenomenon of degradation of the neural response associated with motor activity for healthy aging is known and serves as the background for this research. In addition, our recent studies [47,52] established a reliable increase in the complexity of EEG signals in the elderly subjects and showed a linear correlation between complexity and age. This circumstance is explained by the degeneration of weak neuronal plasticity under the factor of age. Aging has also been shown to lead to a much lower rate of motor initiation in hand clenching, because the motor planning strategies of young and elderly adults differ [48].

We found that elderly subjects demonstrated a stronger increase in DFA scaling exponent during motor tasks, which can be interpreted as more pronounced changes in long-range correlations associated with the transition to “smoother” datasets during the initial parts of EEG data after beginning the hand clenching/unclenching. The rate of increase in α was also higher in the elderly group. However, to avoid significant computing errors when estimating α for short datasets consisting, e.g., of about or less than 100 samples, we applied DFA to EEG segments with a duration of 2 s or 500 samples, which is a quite large amount in relation to the entire duration of an individual motor task. Applying the floating window approach leads to low-pass filtering of α(t), and the resulting dependence becomes fairly smooth when the jumps in α(t) are reduced due to this filtering. As a result, intergroup distinctions become less clear compared to the case of a higher sampling rate with a shorter window duration. Nevertheless, there was significant evidence of stronger responses to motor tasks in electrical activity of the brain in elderly participants.

Another important observation is that the responses in both groups to hand clenching/unclenching are comparable. Although the range of α gain is smaller when unclenched, the relative changes are quite similar. Both of these parts of individual motor tasks can be used to reveal age-related distinctions in the correlation features of EEG data sets with comparable accuracy. Therefore, considering these two parts of each individual task can increase the efficiency of characterizing responses in brain dynamics depending on age.

Although the responses in EEG dynamics caused by clenching the hand by the elderly adults are similar (an increase in the DFA scaling exponent by 22% for the non-dominant hand and by 23% for the dominant hand in Figure 5 and Figure 6 for the C4 channel), the performance of motor tasks by the young group shows different variations in the α value. Figure 5 and Figure 6 demonstrate a similar response for RH, consisting of an increase in α of almost 20% (C4 channel, now this increase is observed over a time interval which is about 0.5 s longer). In the case of LH (non-dominant hand), analogous changes are less pronounced (about 14%, C4 channel). Thus, in addition to the weaker response, there is a larger distinction in scaling exponents between the two hands of young participants compared to elderly adults. Statistical analysis for all EEG channels confirms these distinctions and shows that changes in the correlation features of EEG data caused by motor tasks with dominant and non-dominant hands are relatively strong (about 2.5 times more pronounced than in elderly participants).

An important issue is also the identification of brain regions in which age-related distinctions are clearly defined. According to Figure 7, the most noticeable distinction between the groups of elderly and young adults when performing clenching/unclenching of the non-dominant hand is associated with the area of the sensorimotor cortex (Cz, C4–electrodes).

Currently, a complex network approach is used to characterize the structural and functional systems of the brain [53], and causal relationships between multiple brain regions are examined [54]. It has been established that the sensorimotor cortical system undergoes structural and functional changes throughout its lifespan [55]. Despite the fact that age-related distinctions are established in different areas of the brain, a number of studies confirm the occurrence of such changes in sensorimotor neuromagnetic responses during cued button pressing [56], decleaning of sensorimotor network segregation with age [57], and age effect on automatic inhibitory function of the somatosensory and motor cortex [58].

It should be noted that general conclusions of this study do not depend on the choice of the window size. The latter restricts the range of lg n used to estimate the scaling exponent α. Thus, the data set should contain at least 100 samples to analyze the scales of about lg n = 2.0. Reduced data sets lead to larger statistical errors when fitting the power-law behavior (3). In contrast, data sets of more than 1000 samples provide smoother α(t) dependencies, where short-term responses to motor tasks become less pronounced. A window size of 500 samples is a trade-off that clearly shows changes in EEG signals. We also used other windows, which confirmed the reported age-related distinctions, although the exact values of *r* may decrease with increasing window size.

## 6. Conclusions

Studies on age-related changes in brain electrical activity are important to understand the effects of healthy aging and distinguishing them from pathological changes associated with brain diseases, because even healthy aging can exhibit some of the signs of brain abnormalities. At the latent stages of diseases, distinctions are fairly weak, and their detection is an important problem which requires the application of advanced tools for EEG processing.

We showed that detrended fluctuation analysis of EEG signals provides a useful tool for the assessment of age-related distinctions in the electrical activity of the brain due to the performance of motor tasks by young and elderly adults. The strength of changes in long-range correlations and the rate of increase in DFA scaling exponent are more pronounced in elderly adults. However, the differences in responses to clenching the dominant and non-dominant hands into a fist were better expressed in young participants, who demonstrated relatively weak changes in long-range correlations when performing motor tasks with the non-dominant hand. The potential of DFA can be further used in the field of brain–computer interfaces to recognize different types of real and/or imaginary movements [59] using the EEG signals.

In this study, all participants are right-handed, and this may affect the results. An important question is a comparison of EEG data during execution of motor-related tasks for left-handed young and elderly adults, which could make the conclusion more convincing. This question represent a task for further researches.

## Figures and Tables

**Figure 1 sensors-20-05843-f001:**
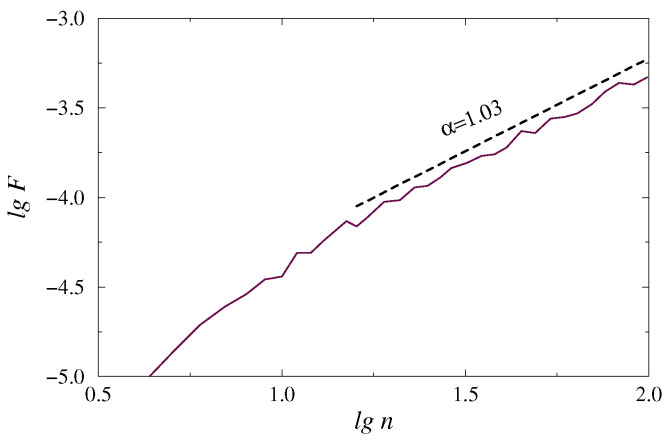
An example of F(n) dependence in the double logarithmic plot for an EEG signal.

**Figure 2 sensors-20-05843-f002:**
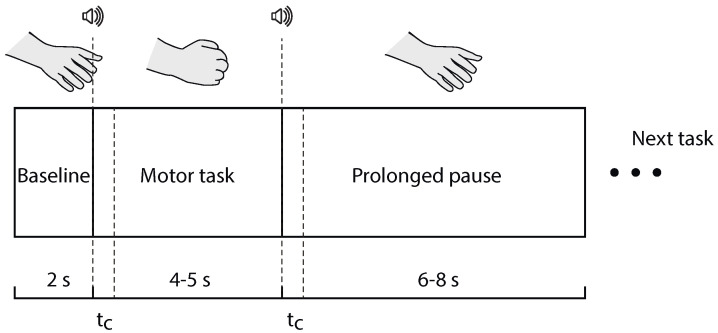
Timeline of an individual motor task. Here, $t_{c}$ denotes the duration of audio signal.

**Figure 3 sensors-20-05843-f003:**
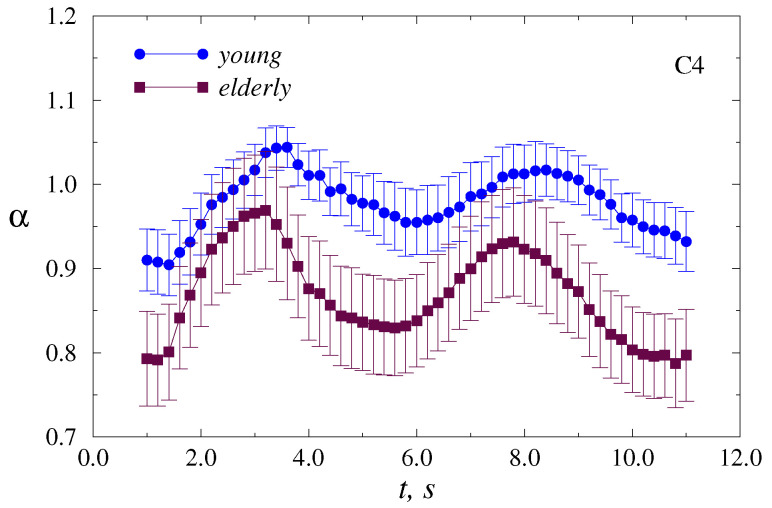
Average dependencies of α on the position of a 2-s (500 samples) floating window, which show the increase in scaling exponent caused by motor tasks. Two segments of increased α are related to clenching/unclenching the hand into a fist.

**Figure 4 sensors-20-05843-f004:**
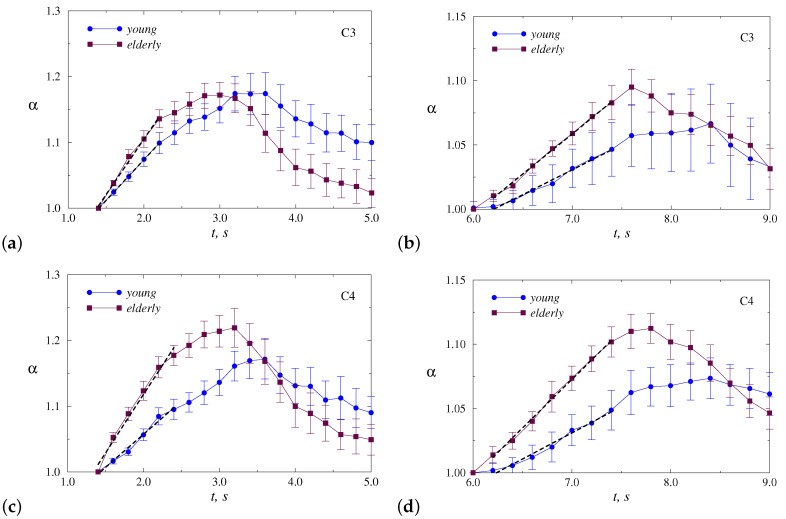
Normalized dependencies α(t) averaged over all motor tasks, which are presented for two segments: clenching (**a**,**c**) and unclenching (**b**,**d**) the hand. Slopes *r* of the dashed lines are used for inter-group separation. The results are shown for the channels C3 (**a**,**b**) and C4 (**c**,**d**).

**Figure 5 sensors-20-05843-f005:**
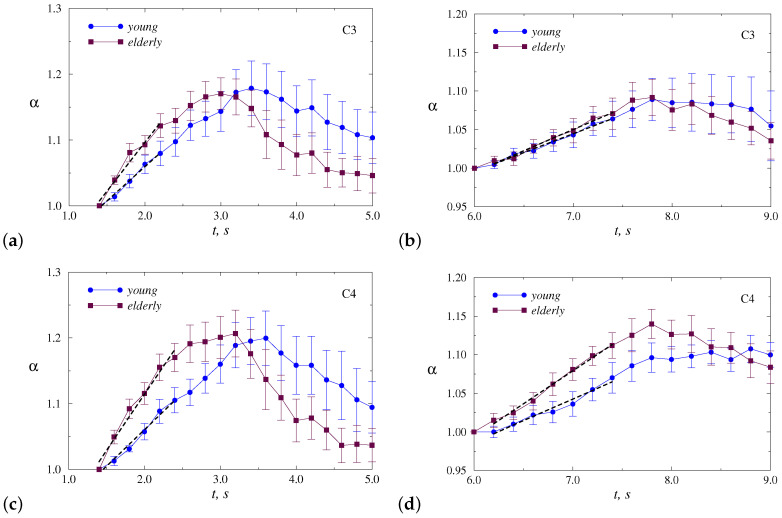
Normalized dependencies α(t) averaged over motor tasks executed by the dominant hand (RH), which are presented for two segments: clenching (**a**,**c**) and unclenching (**b**,**d**) the hand. The results are shown for the channels C3 (**a**,**b**) and C4 (**c**,**d**).

**Figure 6 sensors-20-05843-f006:**
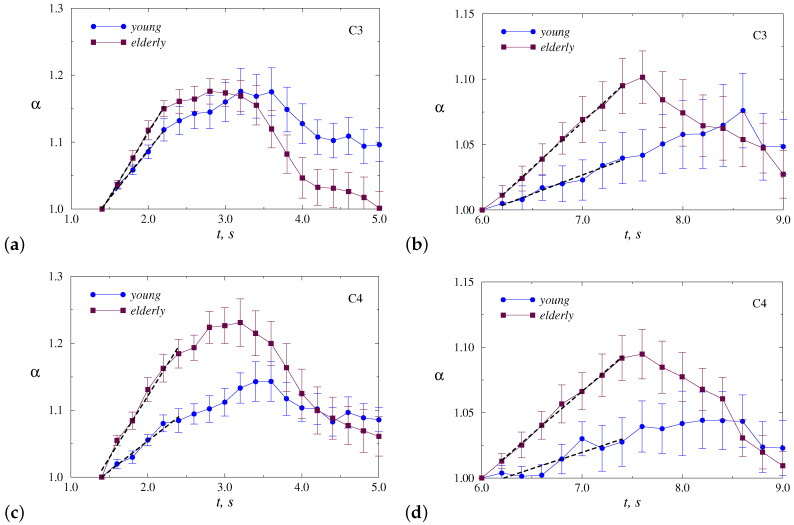
Normalized dependencies α(t) averaged over motor tasks executed by the non-dominant hand (LH), which are presented for two segments: clenching (**a**,**c**) and unclenching (**b**,**d**) the hand. The results are shown for the channels C3 (**a**,**b**) and C4 (**c**,**d**).

**Figure 7 sensors-20-05843-f007:**
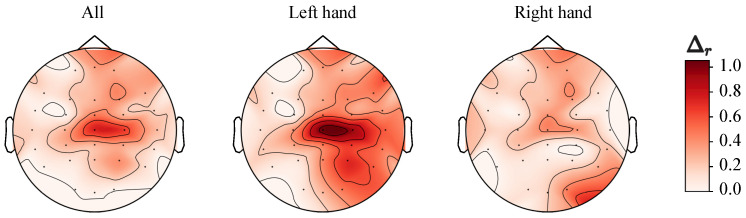
Differences between the *r* values for elderly and young adults depending on the electrode position (normalized).

**Table 1 sensors-20-05843-t001:** Slopes *r* characterizing different rate of growth in α during execution of motor tasks between young and elderly adults. Data are given as mean ± SE.

	All Motor Tasks	Dominant Hand (RH)	Non-Dominant Hand (LH)
*clenching a fist (C3)*			
young	0.095 ± 0.012	0.102 ± 0.011	0.089 ± 0.012
elderly	0.113 ± 0.013	0.107 ± 0.014	0.124 ± 0.011
*clenching a fist (C4)*			
young	0.087 ± 0.013	0.101 ± 0.013	0.072 ± 0.014
elderly	0.121 ± 0.015	0.116 ± 0.016	0.127 ± 0.015
*unclenching a fist (C3)*			
young	0.038 ± 0.008	0.049 ± 0.008	0.026 ± 0.009
elderly	0.054 ± 0.007	0.054 ± 0.006	0.053 ± 0.008
*unclenching a fist (C4)*			
young	0.040 ± 0.009	0.057 ± 0.009	0.024 ± 0.010
elderly	0.068 ± 0.008	0.081 ± 0.008	0.054 ± 0.009

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
