# Peer review of "Age-Related Distinctions in EEG Signals during Execution of Motor Tasks Characterized in Terms of Long-Range Correlations"

_sensors, 2020, doi:10.3390/s20205843_

Round 1

Reviewer 1 Report

Dear Authors,

thank you for submitting your valuable work to Sensors MDPI.

The paper presents DFA as a long range correlation method to explain age-related distinctions in EEG signals during execution of motor tasks.

Please consider following aspects in a revised version of the paper:

  1. include a graphical abstract. 
  2. include the full text of all abbreviations when first used in the text.
  3. include a more detailed mathematical description of the algorithms used.
  4. long range power-law correlations need to be detailed and explained in detail. why are they relevant, what part of the range is relevant (tail) for the study and why.
  5. consider this question: are there neurological reasons why age is affecting these brain structures? what is the neurological substrate of these age related issues? consider the work of Goldberg, E. and Sporns, O. in this matter.
  6. include more details of the case study:
    1. population: i.e. age distribution within the groups, nationality, health status,...
    2. experimental setting.
    3. the case study must be so described that replicability is possible. this is not the case.
  7. expand the conclusions and the implications for the general public.
  8. Link of reference number 44 does not work properly on the PDF. !? 
  9. what is the value for the reader? what are the main takings? please keep this question in mind when re-writing. 

Thank you

Author Response

> 1. include a graphical abstract

We added a graphical abstract.

> 2. include the full text of all abbreviations when first used in the text.

This is done.

> 3. include a more detailed mathematical description of the algorithms used.

We revised the mathematical description and included a new Figure explaining the algorithm (see Section 2, marked by color).

> 4. long range power-law correlations need to be detailed and explained in detail. why are they relevant, what part of the range is relevant (tail) for the study and why.

We added explanations (see Section 2, marked by color).

> 5. consider this question: are there neurological reasons why age is affecting these brain structures? what is the neurological substrate of these age related issues? consider the work of Goldberg, E. and Sporns, O. in this matter.

We added a paragraph at the end of the Discussion with references to other studies of these brain structures (colored).

> 6. include more details of the case study:

>      - population: i.e. age distribution within the groups, nationality, health status,...

>      - experimental setting.

>      - the case study must be so described that replicability is possible. this is not the case.

We extended the description of experiments (Section 3) to include more details of the case study (colored).

> 7. expand the conclusions and the implications for the general public.

This is done.

> 8. Link of reference number 44 does not work properly on the PDF. !? 

This is corrected.

> 9. what is the value for the reader? what are the main takings? please keep this question in mind when re-writing.

We included additional explanations in the Abstract, Discussion, and Conclusions.

Reviewer 2 Report

The authors use window DFA to process EEG signals, and analyze the ability of performing motor tasks in elderly and young people. Although the method DFA is not new, the authors introduce the floating window into DFA, which is easy to use and practical. The three purposes of this paper are clear. The authors achieve these purposes and provide the strong theoretical basis. Besides, the experiment in this paper is comprehensive, and most of the influencing factors have been taken into account in the experiment.

However, the authors mainly use the C4 channel as an example in the paper, because the distinction on the C4 channel is very obvious, but since the experiment is related to the movement of the left and right hands, it is better to provide the specific results of both C3 and C4 channels because these two channels are both closely related to left-right relative tasks. That is also helpful for comparison. Moreover, all testers are right-handed, so this may cause the distribution of the increase in the DFA scaling exponent be biased towards Cz and C4. Will the selection of dominant hand affect the conclusion? Therefore, it should be better to test some EEG data of left-handed young and elderly people, which will make the conclusion more convincing. In addition, how to determine the size of the DFA window? The author did not mention the effect of using other sizes of windows.

Also, in the data collection part, why should the length of sound be used to prompt the participant to perform left-hand or right-hand movement? Why not directly use specific voice prompts or picture prompts.

Author Response

> However, the authors mainly use the C4 channel as an example in the paper, because the distinction > on the C4 channel is very obvious, but since the experiment is related to the movement of the left > and right hands, it is better to provide the specific results of both C3 and C4 channels because these two channels are both closely related to left-right relative tasks. That is also helpful for comparison.

We included Figures 4a,b, 5a,b and 6a,b and additional lines in Table 1 to compare the results of both C3 and C4 channels with the related description.

> Moreover, all testers are right-handed, so this may cause the distribution of the increase in the DFA scaling exponent be biased towards Cz and C4. Will the selection of dominant hand affect the conclusion? Therefore, it should be better to test some EEG data of left-handed young and elderly people, which will make the conclusion more convincing.

Thank you! This is an important question, and we suppose to perform such a comparison in further studies. At present, we do not have a group of left-handed elderly people to obtain reliable results.

> In addition, how to determine the size of the DFA window? The author did not mention the effect > of using other sizes of windows.

We added a paragraph at the end of the Discussion describing this question (colored).

> Also, in the data collection part, why should the length of sound be used to prompt the participant to perform left-hand or right-hand movement? Why not directly use specific voice prompts or picture prompts.

Indeed, presenting visual commands is a wide-spread approach to control the timing and type of executed motor actions in neurophysiological studies. In this context, one of the most accepted strategies is a so-called ‘arrow’ paradigm, proposed in the early works by Pfurtscheller et al. [Pfurtscheller, G., & Neuper, C. (1997). Motor imagery activates primary sensorimotor area in humans. Neuroscience letters, 239(2-3), 65-68.]. In contrast with visual stimulation, audio commands induce greater cortical activation related with much more pronounced launching of sensorimotor integration processes aimed at the external information processing, generating motor command and further control of motor activity [Dushanova, J., & Christov, M. (2014). The effect of aging on EEG brain oscillations related to sensory and sensorimotor functions. Advances in medical sciences, 59(1), 61-67]. The functioning of the emergent perception-action loops is known to be affected by age. As the EEG dataset analysed in this study has been collected as a part of the large-scale studies of healthy ageing [Frolov, N. S. et al. (2020). Age-related slowing down in the motor initiation in elderly adults. PloS one, 15(9), e0233942], we have chosen this particular way to pace the execution of motor action actions in our experimental studies. Besides, audio-pacing is more acceptable than visual stimulation in the experiments with elderly subjects potentially experiencing vision problems.

Round 2

Reviewer 1 Report

Dear Authors,

Thank you for your work and the changes performed.

The paper is, in my opinion, now ready for publication.

All the best,

Javier Villalba-Diez

Reviewer 2 Report

This paper can be accepted in this revised version.